# A Spatial Relationship between Canopy and Understory Leaf Area Index in an Old-Growth Cool-Temperate Deciduous Forest

**Yosuke Tanioka [1], Yihan Cai [1], Hideyuki Ida [2]**  **and Mitsuru Hirota [3],***

[1] Master Degree Program of Mountain Studies, Graduate School of Life and Environmental Sciences, University of Tsukuba, Tsukuba 305-8577, Ibaraki, Japan; s1921266@s.tsukuba.ac.jp (Y.T.); tewnnagi@gmail.com (Y.C.)

[2] Faculty of Education, Shinshu University, Nagano 380-8544, Nagano, Japan; pida@shinshu-u.ac.jp

[3] Faculty of Life and Environmental Sciences, University of Tsukuba, Tsukuba 305-8577, Ibaraki, Japan

\* Correspondence: hirota0313@gmail.com; Tel.: +81-029-853-4753

**Abstract:** Quantification of leaf area index (LAI) is essential for understanding forest productivity and the atmosphere–vegetation interface, where the majority of gas and energy exchange occurs. LAI is one of the most difficult plant variables to adequately quantify, owing to large spatial and temporal variability, and few studies have examined the horizontal and vertical distribution of LAI in forest ecosystems. In this study, we demonstrated the LAI distribution in each layer from the understory to canopy using multiple-point measurements (121 points) and examined the relationships among layers in a cool-temperate deciduous forest. LAI at each point, and the spatial distribution of LAI in each layer, varied within the forest. The spatial distribution of LAI in the upper layer was more heterogeneous than that of LAI at the scale of the entire forest. Significant negative correlations were observed between the upper- and lower-layer LAI. Our results indicate that the understory compensates for gaps in LAI in the upper layer; thus, the LAI of the entire forest tends to remain spatially homogeneous even in a mature forest ecosystem.

**Keywords:** canopy structure; leaf area index; mature forest, understory

## 1. Introduction

To evaluate the productivity of terrestrial ecosystems, it is important to quantify the amount of leaf area and its spatial distribution, as leaves are the only plant organ that can produce organic matter through photosynthesis in nearly all ecosystems. The leaf area index (LAI) is defined as leaf area per unit ground surface area. Previous studies have reported positive correlations between LAI and several variables, such as net primary production, light absorption, microclimate, and water interception [1,2]. Research such as the long-term monitoring of LAI and remote sensing with satellites has been conducted in various ecosystems, and the observed LAI data have been applied to the estimation of productivity [1–6].

Still, understanding of spatial variation in LAI both among and within ecosystems remains limited, due in part to the large quantity of labor involved in estimating LAI. Generally, LAI in an ecosystem is estimated using point-data obtained from litter traps (projected scale: ca. 0.5–1.0 m$^2$), followed by calculations of average LAI per unit area [1,7]. However, the amount of leaf will vary among points even in the same ecosystem depending on the structural heterogeneity at each location. For example, forest heterogeneity can involve factors such as gap formation caused by the death of tall trees via wind or fungal or insect infestations [8,9].

Indirect methods that allow for the point estimation of LAI have been developed [10] and used to report spatial variation in LAI in forests [11–14]. In addition to horizontal heterogeneity, vertical heterogeneity in leaf distribution is another important feature in forest ecosystems. Trees of various heights create a multi-layer structure in forests. Given this multi-layer structure, the spatial distribution of understory LAI may be affected by the layout of leaves of taller trees as well as overall forest structure. However, no studies have investigated both the vertical and horizontal distribution of LAI. If heterogeneously distributed leaves exhibit different physiological and morphological characteristics depending on location, such a structure may lead to errors in the estimation of primary production. Therefore, it is necessary to document both vertical and horizontal heterogeneity of LAI in forest ecosystems.

In this study, we investigated the vertical and horizontal distribution of LAI in a cool-temperature forest. We measured LAI for each vegetation layer at many points using an LAI measurement tool that estimates LAI from the ratio of transmitted near-infrared radiation (NIR) to photosynthetically active radiation (PAR) [15]. This method can measure LAI excluding branches and stems. Several studies have used light detection and ranging (LiDAR) techniques to examine the complexity of forest structure [16,17]. While LiDAR is very useful for detailing forest structure, the method is unable to differentiate between leaves and branches completely, which is necessary to obtain accurate measurements of LAI. In addition to presenting our results, we also discuss how the understory layer compensates for the loss of production in the canopy and which factors control this relationship. In recent studies, LAI was measured through using remote sensing. However, this tends to underestimate LAI in forests with a developed hierarchical structure because remote sensing cannot measure overlapping leaves [6,18]. This study focuses on the hierarchical structure of LAI and attempts to partition the LAI by evaluating each three layers of LAI. This approach is impossible, because it estimates all layers at once. The ground-based measurements of LAI, including LAI partitioning conducted in this study, will contribute to the development of remote sensing of LAI in forests.

## 2. Materials and Methods

### 2.1. Study Site

This study was conducted at the Kayanodaira beech forest in central Japan (36°49′ N, 138°30′ W; 1490 m a.s.l.; Figure 1). The annual mean air temperature was 4.9 °C, and the total annual mean precipitation was 1350 mm. The study site had heavy snowfall from November to May and the maximum depth ranged from 170 to over 400 cm (2003–2020) [19].

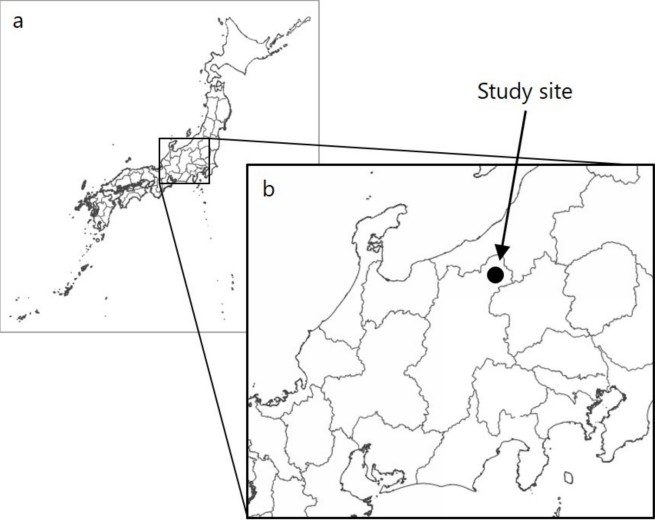

**Figure 1.** Study site (**a**): the map of Japan (**b**): study site.

A 1 ha plot (quadrat of 100 × 100 m) was established in 2005, as a permanent plot of the Japanese Ministry of the Environment's "Monitoring Sites 1000" program [20,21]. The plot was situated in a mature forest dominated by beech (*Fagus crenata*), and the average age of mature beech was over 300 years. In addition to mature beech trees, the forest contained several other deciduous canopy species, such as Birch (Betula ermanii) and Horse chestnut (Aesculus turbinata). The shrub layer consisted of some Maples (Acer nipponicum, A. japonicum, and A. rufinerve), and various deciduous shrubs (Hydrangea paniculata, Viburnum furcatum, etc.). Data for forest inventory were provided by the Ministry of the Environment Monitoring Sites 1000 Project (SIN01.zip, Fujiyoshida, Japan) (Table 1). The forest floor was covered with a dense understory of *Sasa senanensis*, an evergreen perennial rhizomatous dwarf-bamboo. The maximum height of the *S. senanensis* thicket was ca. 1.5 m. The survey area had a well-developed gap mosaic structure (Figure 2), with a large number of shrubs in gaps, and a small number of shrubs under a closed canopy.

**Table 1.** Summary of stand characteristics.

| Species | Trees (No/ha) | Basal Area [1] (m²/ha) | Relative Trees (%) | Relative Basal Area (%) | Average DBH [2] (cm) |
|---|---|---|---|---|---|
| *Fagus crenata* | 220 | 27.250 | 23.305 | 82.510 | 29.797 |
| *Betula ermanii* | 5 | 1.168 | 0.530 | 3.535 | 48.020 |
| *Aesculus turbinata* | 17 | 0.924 | 1.801 | 2.799 | 22.237 |
| *Acer nipponicum* | 183 | 0.914 | 19.386 | 2.767 | 7.612 |
| *Hydrangea paniculata* | 132 | 0.521 | 13.983 | 1.577 | 6.945 |
| *Chengiopanax sciadophylloides* | 22 | 0.502 | 2.331 | 1.519 | 15.066 |
| *Acer japonicum* | 74 | 0.435 | 7.839 | 1.316 | 8.182 |
| *Viburnum furcatum* | 87 | 0.266 | 9.216 | 0.804 | 6.166 |
| *Phellodendron amurense* | 10 | 0.189 | 1.059 | 0.573 | 13.920 |
| *Sorbus commixta* | 34 | 0.173 | 3.602 | 0.523 | 7.712 |
| *Cornus controversa* | 54 | 0.166 | 5.720 | 0.504 | 6.205 |
| *Padus grayana* | 21 | 0.100 | 2.225 | 0.303 | 7.618 |
| *Acer pictum* | 1 | 0.095 | 0.106 | 0.287 | 34.728 |
| *Euonymus macropterus* | 18 | 0.074 | 1.907 | 0.223 | 7.075 |
| *Acer rufinerve* | 11 | 0.069 | 1.165 | 0.210 | 8.826 |
| *Corylus sieboldiana* | 23 | 0.057 | 2.436 | 0.173 | 5.591 |
| *Tilia japonica* | 7 | 0.057 | 0.742 | 0.171 | 9.572 |
| *Symplocos sawafutagi* | 20 | 0.052 | 2.119 | 0.158 | 5.723 |
| *Acer tschonoskii* | 2 | 0.008 | 0.212 | 0.025 | 7.257 |
| *Toxicodendron trichocarpum* | 3 | 0.008 | 0.318 | 0.025 | 5.931 |
| Total | 944 | 33.027 | | | |

[1] Basal Area; The cross-sectional area of trees at breast height. [2] The Diameter of trees at Breast Height.

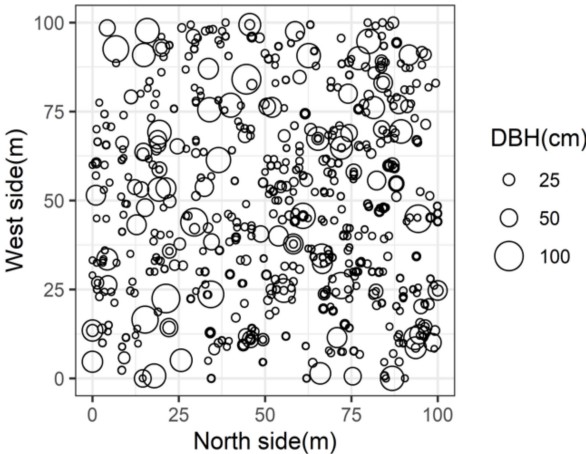

**Figure 2.** The spatial distribution of living trees. The circle size indicates diameter of breast height (DBH) of living trees their DBH are ≥ 5 cm.

## 2.2. Leaf Area Index

We used a portable leaf area index analyzer (MIJ-LAI/P; Environmental Measurement Japan, CO., LTD., Fukuoka, Japan) to estimate LAI. This device estimates LAI from the ratio of transmitted near-infrared radiation (NIR) to photosynthetically active radiation (PAR) using Equation (1) [15]

$$\text{LAI} = 2.8 \times \ln(\text{NIR/PAR}) + 0.69 \tag{1}$$

PAR is measured at wavelengths from 400 to 700 nm, while NIR is measured from 700 to 1000 nm. If many leaves cover the sensor, the amount of PAR will be small but the amount of NIR will not change; therefore, the LAI value will be high. Values of LAI measured using this device are only affected by green leaves. Kume et al. (2011) showed that the NIR/PAR correlated with the seasonal variation in LAI and NDVI, and stated that NDVI responded to the different reflectance characteristics of the leaves that may affect LAI. They measured LAI using this method in a deciduous broadleaf forest. Therefore, we adopted this method in this study site that is similar to the site of Kume et al. (2011).

Measurements of LAI were conducted for 2 days in August 2018, and 7 days in August 2019. In 2018, the measurements were conducted from 9 a.m. to 4 p.m., since it was cloudy all the days. Meanwhile, in 2019, LAI measurement was conducted before sunrise and after sunset since it was a clear day. The main reason for avoiding LAI measurement during daytime was because direct light caused errors in LAI estimation using this device. Furthermore, LAI measurement was not taken when it was raining because dense fog also causes errors (Kume et al. (2011)). We had rain in one afternoon during te study period. LAI peaked from early July to early October at this site (data not shown). We systematically measured LAI in the 1 ha study plot by establishing 121 points at which to measure LAI, using 10 m grid lines. Then, LAI was measured at different heights (0 and 2.5 m above the ground in 2018; 0, 2.5, and 5 m above the ground in 2019) at each point. LAI measured at 5 m above the ground ($\text{LAI}_5$) was assumed to represent the LAI of the canopy layer; that measured at 2.5 m above the ground ($\text{LAI}_{2.5}$) was considered to represent all layers excluding the understory dwarf-bamboo layer; and that measured at 0 m above the ground ($\text{LAI}_0$) represented that of all three layers. In addition, we calculated three types of LAI (Figure 3): the understory, dwarf bamboo, and shrub layers, using two different LAI values, as seen in Equations (2)–(4) corresponding to

$$\text{Understory-layer (LAI}_U\text{): } \text{LAI}_U = \text{LAI}_0 - \text{LAI}_5 \tag{2}$$

$$\text{Dwarf bamboo-layer (LAI}_D\text{): } \text{LAI}_D = \text{LAI}_0 - \text{LAI}_{2.5} \tag{3}$$

$$\text{Shrub-layer (LAI}_S\text{): } \text{LAI}_S = \text{LAI}_{2.5} - \text{LAI}_5 \tag{4}$$

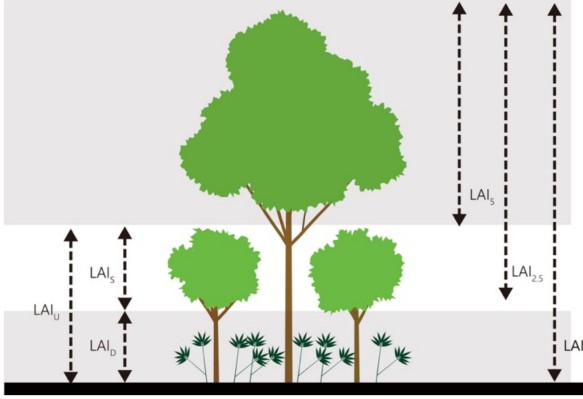

**Figure 3.** The design of leaf area index (LAI) measurement. Understory-layer ($\text{LAI}_U$); Dwarf bamboo-layer ($\text{LAI}_D$); Shrub-layer ($\text{LAI}_S$); LAI measured at 5 m above the ground ($\text{LAI}_5$); LAI measured at 2.5 m above the ground ($\text{LAI}_{2.5}$); LAI measured at 0 m above the ground ($\text{LAI}_0$).

To eliminate noise caused by sun flex [15], measurements were only conducted immediately before sunrise and after sunset, or during cloudy periods. In 2019, LAI measurements were repeated three times, and the average was considered the LAI at each grid point. The measurements were not repeated in 2018.

## 3. Results

### 3.1. Spatial Distribution of LAI

The LAI of the lower layer ($LAI_{2.5}$) accounted for approximately one-third of the total LAI ($LAI_0$) in both years (Table 2). The averages ± Standard Deviation (SD) of $LAI_{2.5}$ in 2018 and 2019 were 2.41 ± 0.725 and 2.17 ± 0.782, respectively. Values for $LAI_0$ in 2018 and 2019 were 3.66 ± 0.425 and 3.01 ± 0.471, respectively (Table 2). No significant differences were observed between $LAI_{2.5}$ and $LAI_5$ in 2019 ($p = 0.21$). The Coefficient of Variation (CV) of upper-layer LAI became larger than that of lower-layer LAI (Table 2). Hence, the spatial distribution of LAI of the upper layers was more heterogeneous than that of the lower layers. Compared to the range of LAI observed in previous studies (2.5–5.9), the value of LAI in this study was low (Table 3).

**Table 2.** LAI summary (n = 121).

| Year | Layer | Average (LAI) | SE [1] (LAI) | CV (LAI) |
|---|---|---|---|---|
| **2018** | $LAI_0$ | 3.66 | 0.0386 | 11.6 |
| | $LAI_{2.5}$ | 2.41 | 0.0659 | 31.1 |
| **2019** | $LAI_0$ | 3.01 | 0.0428 | 15.7 |
| | $LAI_{2.5}$ | 2.17 | 0.0711 | 36.0 |
| | $LAI_5$ | 2.00 | 0.0828 | 45.6 |

[1] Standard Error; The Coefficient of Variation (CV); LAI measured at 5 m above the ground ($LAI_5$); LAI measured at 2.5 m above the ground ($LAI_{2.5}$); LAI measured at 0 m above the ground ($LAI_0$).

**Table 3.** LAI value of some deciduous broadleaf forests.

| Study | Site (Region) | Method or Tool | Time | LAI (±SE) | Dominant Species |
|---|---|---|---|---|---|
| This study | Kayanodaira (Nagano, Japan) | NIR/PAR ratio | August 2018 | 2.5 (±0.039) | *Fagus crenata* |
| This study | Kayanodaira (Nagano, Japan) | NIR/PAR ratio | August 2019 | 2.2 (±0.082) | *Fagus crenata* |
| Kume et al., 2011 [15] | Takayama (Gifu, Japan) | NIR/PAR ratio | June 2006 | 5.1 | *Betula ermanii, Quercus crispula* |
| Nasahara et al., 2008 [3] | Takayama (Gifu, Japan) | PAR transmittance | 2005~2006 | 5.1~5.9 | *Betula ermanii, Quercus crispula* |
| Nasahara et al., 2008 [3] | Takayama (Gifu, Japan) | Litter fall | 2005~2006 | 5.0 | *Betula ermanii, Quercus crispula* |
| Nasahara et al., 2008 [3] | Takayama (Gifu, Japan) | LAI-2000 [1] | 2005~2006 | 3.0 | *Betula ermanii, Quercus crispula* |
| Melnikova et al., 2018 [22] | Takayama (Gifu, Japan) | PAR transmittance | May~August 2013 | 5.9 | *Betula ermanii, Quercus crispula* |
| Melnikova et al., 2018 [22] | Takayama (Gifu, Japan) | Litter fall | May~August 2013 | 5.0 | *Betula ermanii, Quercus crispula* |
| Melnikova et al., 2018 [22] | Takayama (Gifu, Japan) | remote sensing by satellite | May~August 2013 | 5.5 | *Betula ermanii, Quercus crispula* |
| Bequet et al., 2012 [11] | Flanders (Belgium) | Hemispherical photographs | August 2008 | 2.5~3.3 | *Fagus sylvatica Quercus robur* |
| Granier et al., 2008 [23] | Hesse forest (north-eastern France) | Litter fall | 1996~2005 | 4.6~7.2 | *Fagus sylvatica* |
| Ngao et al., 2011 [24] | Hesse forest (north-eastern France) | LAI-2000 [1] | 2004 | 4~8.1 | *Fagus sylvatica* |
| Cerny et al., 2020 [25] | Training Forest Enterprise Masaryk Forest (Křtiny, Czech) | Litter fall | 2013 | 5.2~5.6 | *Fagus sylvatica* |
| Cerny et al., 2020 [25] | Training Forest Enterprise Masaryk Forest (Křtiny, Czech) | Needle Technique | 2013 | 3.4~6.0 | *Fagus sylvatica* |
| Cerny et al., 2020 [25] | Training Forest Enterprise Masaryk Forest (Křtiny, Czech) | LAI-2000 [1] | 2013 | 4.5~5.1 | *Fagus sylvatica* |
| Glatthorn et al., 2018 [26] | eastern, Slovakia | LAI-2000 [1] | 2013 | 6.2 (±0.39) | *Fagus sylvatica* |
| Glatthorn et al., 2018 [26] | eastern, Slovakia | Litter fall | 2013 | 8.5 (±0.54) | *Fagus sylvatica* |
| Asner et al., 2003 [2] | Various | Various | Various | 5.1 (±0.13) | Various |

[1] LAI-2000 Plant Canopy Analyzer (Li-COR, Lincoln, NE, USA); near-infrared radiation (NIR); photosynthetically active radiation (PAR).

Analyses of contour maps indicated that LAI at each layer, $LAI_0$, $LAI_{2.5}$, and $LAI_5$, varied greatly in the horizontal direction at the study plot (Figure 4). In addition, the spatial pattern of the contour map at the same layer, particularly in $LAI_{2.5}$, differed between 2018 and 2019 (Figure 4b,d). A large gap was present at grid point $40 \times 70$ m, located slightly above the center of the study plot in 2018; however, the gap had filled with leaves in 2019 (Figure 4b,d).

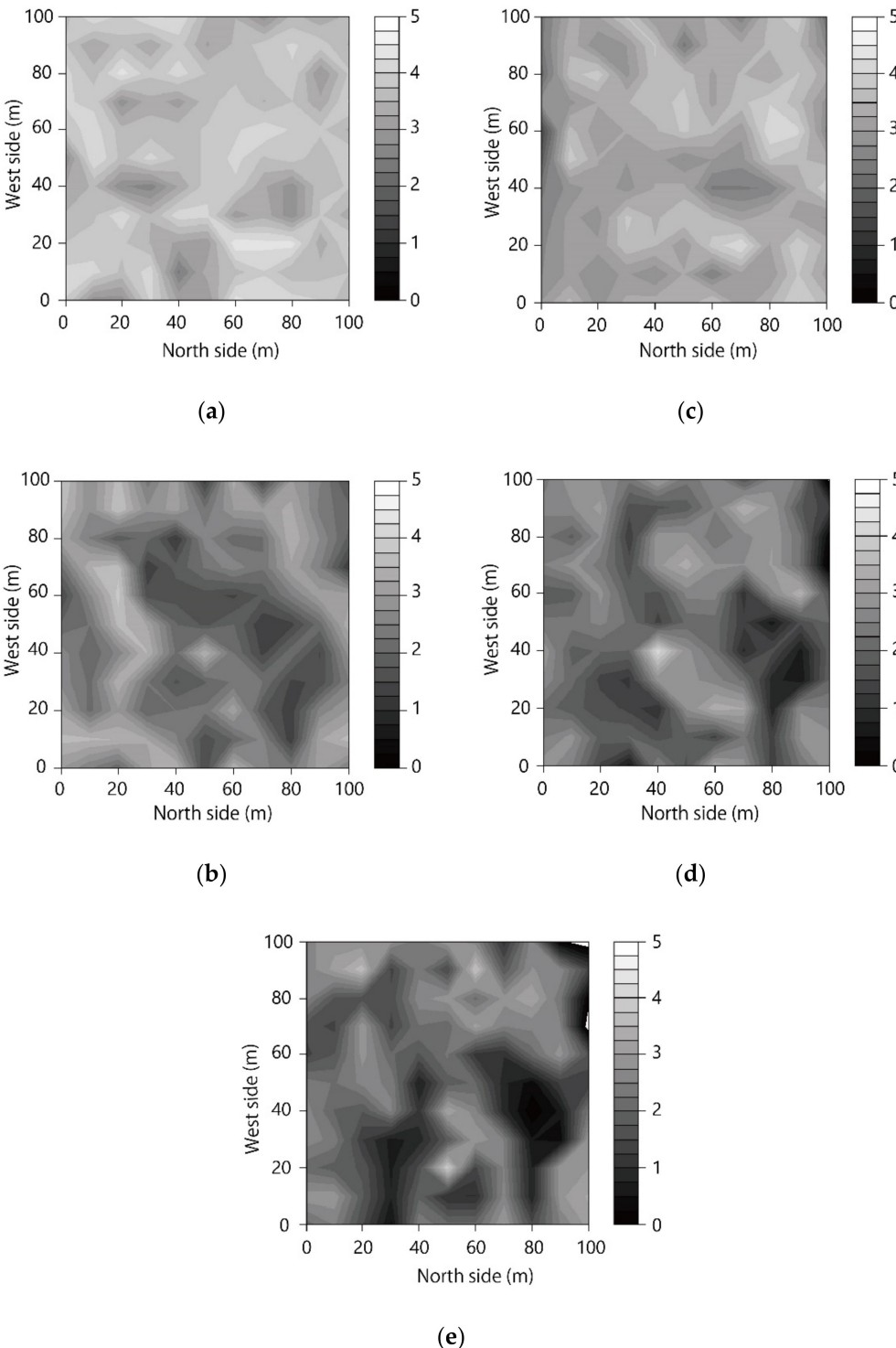

**Figure 4.** Spatial distribution of LAI. (**a**): $LAI_0$ in 2018. (**b**): $LAI_{2.5}$ in 2018. (**c**): $LAI_0$ in 2019. (**d**): $LAI_{2.5}$ in 2019. (**e**): $LAI_5$ in 2019. The axis is coordinate in study site. The contour indicates LAI.

## 3.2. The Relationship among Layers

Negative correlations were observed between values of LAI in the lower and higher layers (Figure 5). The correlation coefficients between $LAI_{2.5}$ and $LAI_D$ in 2018 and 2019 were $-0.847$ ($p < 0.001$) and $-0.765$ ($p < 0.001$), respectively. The correlation coefficients between $LAI_5$ and $LAI_U$, $LAI_5$ and $LAI_S$, and $LAI_5$ and $LAI_D$ were $-0.851$ ($p < 0.001$), $-0.574$ ($p < 0.001$), and $-0.640$ ($p < 0.001$), respectively. These results indicate that the LAI of the lower layer is distributed in a complementary manner to the LAI of the upper layer. No significant correlation was observed between $LAI_S$ and $LAI_D$ ($r = 0.021$, $P = 0.818$) (Figure 5e), indicating that $LAI_S$ and $LAI_D$ each independently compensate for canopy gaps.

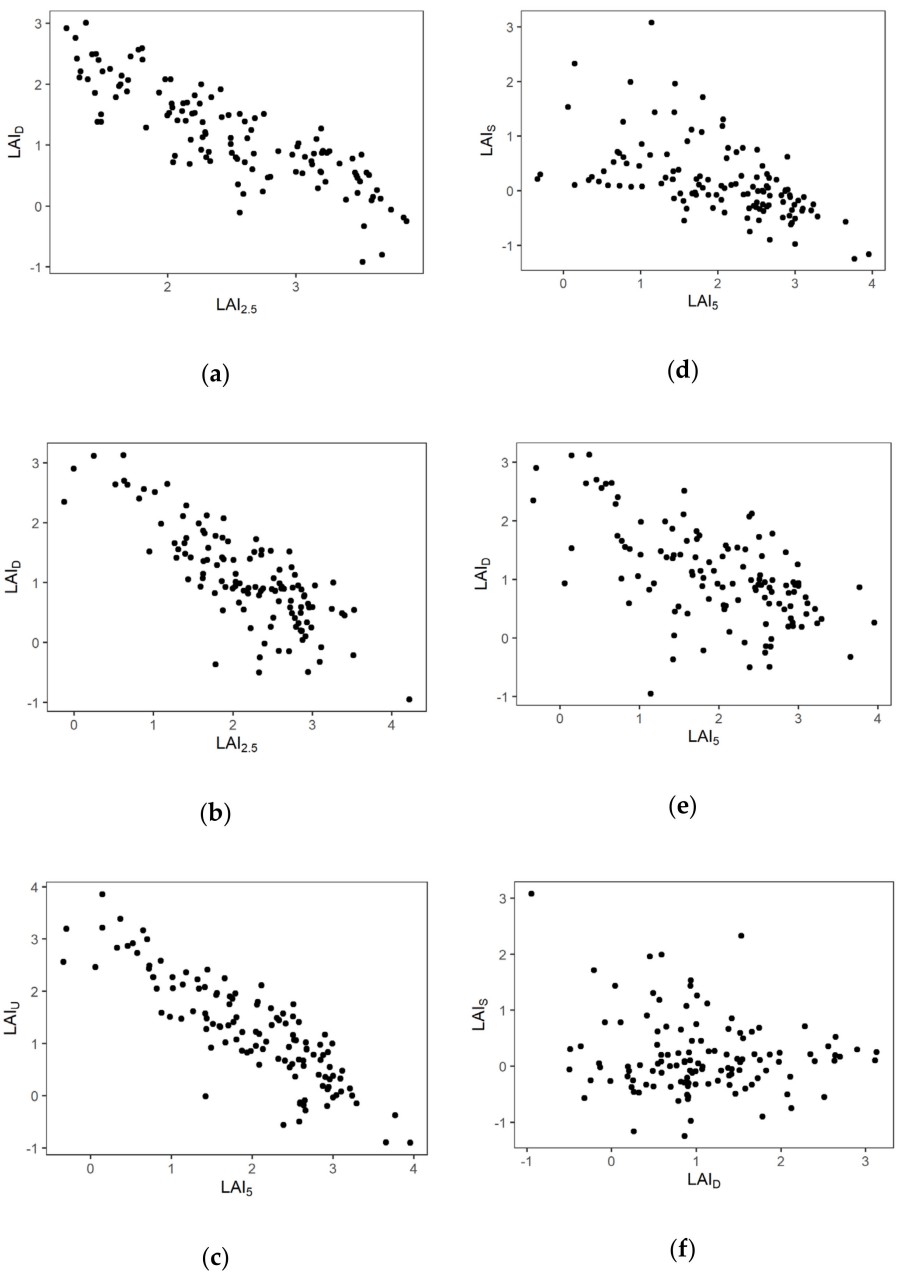

**Figure 5.** Relationship between layers. (**a**): $LAI_{2.5}$ and $LAI_D$ in 2018. (**b**): $LAI_{2.5}$ and $LAI_D$ in 2019. (**c**): $LAI_5$ and $LAI_U$. (**d**): $LAI_5$ and $LAI_S$. (**e**): $LAI_5$ and $LAI_D$. (**f**): $LAI_D$ and $LAI_S$ One point indicates one grid point.

## 4. Discussion

### 4.1. Spatial Distribution of LAI and Relationship among Layers

LAI in the upper layers ($LAI_{2.5}$ and $LAI_5$) was heterogeneous at a 10 m scale even within the same forest (Table 2, Figure 4). Several previous studies have investigated spatial variation in LAI, and the magnitude of variation has differed among studies. At the 10 m scale in 10 beech forests, Bequet et al. (2012) reported that the CV of LAI ranged from 7.0 to 23.9 [11]. At the 1.5 m scale along 15 m transects, Glatthorn et al. (2017) found that the CV of LAI ranged from 28.1 to 31.0 among three forests [26]. Liu et al. (2018) reported a CV of 21 for the LAI distribution at a 30 m scale [14]. Because both the scale and measurement methods differ among these studies (and the present study), it may not be appropriate to directly compare CV values. However, the CV of $LAI_{2.5}$, which is the general height used in LAI research, was higher than these previously reported values (Table 2). This difference is likely due to the distinct gap mosaic structure in our study forest. In the LAI contour maps, we were able to detect several clear canopy gaps that exhibited lower or zero values of LAI, e.g., at $40 \times 30$ m, $40 \times 60$ m, and $80 \times 10$ m (Figure 4). During LAI measurement, we found one to several dead, fallen trees within each canopy gap. Strong disturbances such as typhoons and heavy snowfall contribute to canopy gaps in East Asia [27]; thus, such events likely contributed to the formation of the large gap area at the study site.

Our results indicate that the leaves of the understory can compensate for the lack of leaf area at the canopy layer. $LAI_0$ was distributed homogeneously under conditions of a heterogeneously distributed upper-layer LAI (Figure 4). In fact, we observed significant negative correlations between upper-layer and lower-layer LAI, consistent with Majasalmi et al. (2020) [28]. Similar complementary effects of a multi-layered LAI in forest ecosystems have been suggested to stabilize levels of primary production in old-growth forests with a gap mosaic and multi-layer structure [4,28–32]. Considering the substantial proportion of total LAI accounted for by lower-layer (understory) LAI, this compensatory effect is of great importance. For lower and upper LAI (separated at 4.5 m height) in a European beech forest, Glatthorn et al. (2017) reported that the fraction of lower LAI out of total LAI was approximately 0.15 [26]. In a cool-temperate deciduous forest dominated by dwarf bamboo, similar to our study forest, Nasahara et al. (2008) estimated this fraction to be approximately 0.2–0.3 [3], comparable to the value observed in the present study. Although further research is needed, the value of this fraction of LAI in forest ecosystems dominated by dwarf bamboo may be largely due to the ecological traits of this species, an evergreen and vigorously rhizomatous perennial.

The dwarf bamboo and shrubs under the canopy layer both play key compensatory effects, but the extent of those effects is likely to differ between the two types of vegetation. Our observed relationships among layers suggest that the LAI of the shrub layer was more sensitive to the upper light environment determined by canopy structure than was dwarf bamboo. Thus, the $LAI_U$–$LAI_5$ relationship had higher regression coefficients compared to the $LAI_D$–$LAI_{2.5}$ relationship in 2019 (Figure 5). This difference may reflect the variation in the growth forms of the two species, as shrubs exist as distinct individuals, while dwarf bamboo is a clonal plant.

### 4.2. Rapid Changes in LAI Estimated from 2 Years of Measurements

The spatial pattern of LAI differed between 2018 and 2019. The value of upper-layer LAI and the size and distribution within the canopy gap appeared to change, particularly for $LAI_{2.5}$. At some point locations, LAI was increasing, while at others, values were decreasing (Figure 5). The increase in LAI in just 1 year would indicate that the canopy beech trees surrounding gaps extended their branches and contributed to filling in the gap. Feldmann et al. (2018) reported that the horizontal elongation of the branches of tall trees is the primary manner by which small canopy gaps are filled [32]. Two processes generally contribute to the filling of canopy gaps in forest ecosystems: the horizontal development of tree branches and the vertical development of regeneration layers. However, the vertical development of regeneration layers would have been unlikely to contribute to the rapid changes in LAI observed

here, because growth of the regeneration layer appeared to be slow at our study site. Meanwhile, the decrease in LAI in just 1 year would have been caused by fallen and/or broken live trees or large branches near the canopy layer. This process is more likely to occur in mature forests, such as our study site, than in younger forests.

*4.3. Future Task*

Since our primary objective was to clarify the relative spatial and vertical variations in LAI within the forest, the accuracy of the LAI value was not taken into consideration in this study. However, our data showed that the LAI values in our study were relatively low compared with other studies (Table 3), probably due to a mismatch in the coefficients in the estimation equation (Equation (1)) for this study site. We could not verify whether this method underestimates LAI or not because we did not know the true LAI in our site. However, If LAI was really underestimated, it is expected that our result was slightly affected quantitatively by underestimation of LAI. Correctly estimating the LAI using this simple method will require to correct for the coefficients for the comparison of the actual observations with the estimates.

## 5. Conclusions

We investigated the spatial distribution of LAI for several vegetation layers in a mature beech forest. The distribution differed among layers, and negative correlations were observed between the lower and upper layers. The LAI of the canopy was distributed heterogeneously both horizontally and vertically, while that of the understory was distributed in a manner that compensated for the loss of LAI in canopy gaps.

**Author Contributions:** Conceptualization, Y.T. and M.H.; methodology, Y.T.; formal analysis, Y.T.; investigation, Y.T., Y.C. and M.H.; resources, H.I.; data curation, Y.T.; writing—original draft preparation, Y.T.; writing—review and editing, M.H.; supervision, M.H.; project administration, Y.T. and M.H.; funding acquisition, M.H. All authors have read and agreed to the published version of the manuscript.

**Funding:** This study was supported in part by the Master's Degree Program of the Department of Mountain Studies, University of Tsukuba, and by Grants-in-Aid for Scientific Research from the Japan Society for the Promotion of Science (JSPS; No. 17KT0068).

**Acknowledgments:** We thank Koyama at the Graduate University for Advanced Studies, SOKENDAI, and Seki at the University of Tsukuba for field research assistance.

**Conflicts of Interest:** The authors declare no conflict of interest.

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
