# Peer review of "A Spatial Relationship between Canopy and Understory Leaf Area Index in an Old-Growth Cool-Temperate Deciduous Forest"

_forests, doi:10.3390/f11101037_

Round 1

Reviewer 1 Report

The introduction needs to elaborate in the topics of methods and applications of LAI. readers may want to know the difference between readings taken from the ground and their relationship with those derived from satellite or RS images. how comparisons from insitu are match to pixel size readings...

in the methods, the paper must include detail information regarding the equipment itself examples of other similar experiments using it and the requirements for calibration or potential sources of error. is any difference between broadleaf or evergreen trees when sampling...

Also, it needs to describe in detail the sampling process, how many samples were collected per site, how many days were used to collect samples. if sunlight intensity is source of error how they assure similarities between first and last reading on a sampling day.  are any other variables potentially inducing error. rain fall, drought, hydration of the plans...

Author Response

Reviewer 1 Comments and Suggestions

1)  The introduction needs to elaborate in the topics of methods and applications of LAI. readers may want to know the difference between readings taken from the ground and their relationship with those derived from satellite or RS images. how comparisons from insitu are match to pixel size readings...

We totally agree with your supportive comments and deem that it is essential that relationship between our study and recent mainstream LAI researches using remote sensing information. To emphasize the relationship and importance of our study, we have added the following sentences in the section of Introduction (P1, L61 – L67.).

In recent studies, LAI measured through using remote sensing. However, it tends to underestimate LAI in forests with developed hierarchical structure because remote sensing cannot measure leaves overlapping [6, 19]. This study focuses on the hierarchical structure of LAI and attempts to partition the LAI by evaluating each three layers of LAI. It is impossible with remote sensing because it estimates all layers at once. The ground-based measurements of LAI, including LAI partitioning conducted in this study, will contribute to the development of remote sensing of LAI in forests.

2)  in the methods, the paper must include detail information regarding the equipment itself examples of other similar experiments using it and the requirements for calibration or potential sources of error. is any difference between broadleaf or evergreen trees when sampling...

Maybe you know, the equipment used in this study was released in 2015 and there are still no other previous studies except for Kume et al. (2011). As we already mentioned that Kume et al. (2011) reported that the estimated LAI values by using this equipment was strongly correlated with observed LAI in a deciduous broadleaf forest in Japanese cool-temperate region. For the main objectives of this study, to demonstrate the vertical and horizontal distribution of LAI in a cool-temperature forest, we believe that the estimated LAI values by using this equipment will be able to provide a sound basis for our study. But, we added some descriptions by referring equipment instruction flowing your advice as follows (P4, L100 – L103.).

Kume et al (2011) showed that the NIR/PAR well correlated with seasonal variation of LAI and NDVI, and stated that NDVI responded to the different reflectance characteristics of the leaves that may affect to LAI. They measured LAI using this method in a deciduous broadleaf forest. Therefore, we adopted this method in this study site that is similar to the site of Kume et al. (2011).

3)  Also, it needs to describe in detail the sampling process, how many samples were collected per site, how many days were used to collect samples. if sunlight intensity is source of error how they assure similarities between first and last reading on a sampling day.  are any other variables potentially inducing error. rain fall, drought, hydration of the plans...

As you suggested, we have revised explanation about sampling LAI at the beginning in the section of LAI measurement (P4, L104 – L110.), as follows.

Measurements of LAI were conducted for 2 days in August 2018, 7 days in August 2019, respectively. As for 2018, the measurements were conducted from 9 a.m. to 4 p.m. since it was cloudy all the day. Meanwhile in 2019, LAI measurement was conducted before sunrise and after sunset since it was clear day. Main reason for avoiding LAI measurement during daytime was highly diffused transmitted light and cause errors for LAI estimation by using the device. Furthermore, LAI measurement were not taken when it's raining or thick clouds measurements because dense fog and raindrops also cause errors (Kume et al. (2011)).

Reviewer 2 Report

Line 37. “m2” (the 2 should be a superscript).

Lines 66–67. “The study site was covered by 3–4 m of snow …”. Do you mean in 2018, when measurements were made, or is this a long term average, in which case the text needs modification?

Lines 84–85. “Both axes indicate the coordinate …”. “The positions … indicate the coordinates …” There is something funny here; rewrite.

Line 119, Table 2. In 2018, shouldn’t the entry be 31.1 (=100*0.725/2.41) instead of 28.9? Also, the heading of 4th column should be SD, not Sd.

Line 120, Table 3. Title: should be “forests”, not “forest”. Also, the heading of the 5th column should be LAI(±SD) not LAI(±s.d.). However, use of SD in Tables 2 & 3 is not helpful for deciding whether LAI differs between years and between studies. SE (standard error) is much better for this purpose. This reviewer recommends that SE be used rather than SD for the entries in both tables.

Line 120, Table 3. The last entry in the body of the table is “not description”. That should read “not described”.

Line 175. Shouldn’t that read “dwarf bamboo is a clone” (not the clone)?

Lines 191–196. This is a bit worrisome. Publishing the paper as it stands will result in the paper being referenced and values of LAI quoted by subsequent publications. If these values are unreasonably low (due to an inadequacy in the estimation equation 1), the risk is that incorrect values of LAI will be perpetuated. Perhaps the authors should add a sentence to this “Future task” section to emphasize that even if LAI values are low, they tend to be low throughout, and that may have a minimal effect on conclusions that are made involving differences between LAI values.

Line 207. “please turn to the CRediT taxonomy …” Something is wrong here!

Author Response

Reviewer 2 Comments and Suggestions

Line 37. “m2” (the 2 should be a superscript).

Thank you for noticing, and we have changed it (P1, L37.).

Lines 66–67. “The study site was covered by 3–4 m of snow …”. Do you mean in 2018, when measurements were made, or is this a long term average, in which case the text needs modification?

It is a long term average. We have revised explanation as follows (P2, L72 – L73.).

The study site have had heavy snowfall from November to May that the maximum depth was ranged from 170 cm to over 400 cm (2003 – 2020) [20].

Lines 84–85. “Both axes indicate the coordinate …”. “The positions … indicate the coordinates …” There is something funny here; rewrite.

We have revised as follows (P4, L91 – L92.).

The spatial distribution of living trees. The circle size indicates diameter of breast height (DBH) of living trees their DBH are >=5 cm.

Line 119, Table 2. In 2018, shouldn’t the entry be 31.1 (=100*0.725/2.41) instead of 28.9? Also, the heading of 4th column should be SD, not Sd.

We apologize our careless miss and have done accordingly, as follows (P5, L134 – L135.). As for “SD”, we have changed to “SE” as your appropriate next indication.

”28.9” → ”31.1” and ”Sd” → ”SE”

Line 120, Table 3. Title: should be “forests”, not “forest”. Also, the heading of the 5th column should be LAI(±SD) not LAI(±s.d.). However, use of SD in Tables 2 & 3 is not helpful for deciding whether LAI differs between years and between studies. SE (standard error) is much better for this purpose. This reviewer recommends that SE be used rather than SD for the entries in both tables.

Thank you for your fair suggestion. We have changed Table 2 as your suggestion (P5, L134 – L135.). However, we couldn’t get SE about some previous studies in table 3, so we have changed only a part of table 3 (P4 - P5, L135 – L136.) as follows;

         "LAI (± SD)” → “LAI (± SE)”

         ” 5.9(± 0.4)” → ”5.9”

         ” 5.0(± 0.4)” → ”5.0”

         ” 5.5(± 0.2)” → ”5.5”

         ” 6.2(± 1.8)” → ”6.2(±0.39)”

         ” 8.5(± 2.5)” → ”8.5(±0.54)”

         ” 5.1(± 1.8)” → ”5.1(±0.13)”

Line 120, Table 3. The last entry in the body of the table is “not description”. That should read “not described”.

Thank you for your suggestion. but we thought that it should read “Various” because the paper is review and it treated various species. Therefore, we have revised it.  (P4 - P5, L135 – L136.).

      “not description” → “Various”

Line 175. Shouldn’t that read “dwarf bamboo is a clone” (not the clone)?

We have revised it as follows (P9, L190.).

dwarf bamboo is a clonal plant.

Lines 191–196. This is a bit worrisome. Publishing the paper as it stands will result in the paper being referenced and values of LAI quoted by subsequent publications. If these values are unreasonably low (due to an inadequacy in the estimation equation 1), the risk is that incorrect values of LAI will be perpetuated. Perhaps the authors should add a sentence to this “Future task” section to emphasize that even if LAI values are low, they tend to be low throughout, and that may have a minimal effect on conclusions that are made involving differences between LAI values.

Thank you very much for your supportive suggestions. We have added some sentences as follows (P9, L210 – L211.).

If LAI was really underestimated, it’s expected that our result was slightly affected quantitatively by underestimation of LAI.

Line 207. “please turn to the CRediT taxonomy …” Something is wrong here!

We have deleted it.
